# Potent Apoptosis Induction by a Novel Trispecific B7-H3xCD16xTIGIT 2+1 Common Light Chain Natural Killer Cell Engager

**DOI:** 10.3390/molecules29051140

**Published:** 2024-03-04

**Authors:** Michael Ulitzka, Julia Harwardt, Britta Lipinski, Hue Tran, Björn Hock, Harald Kolmar

**Affiliations:** 1Institute for Organic Chemistry and Biochemistry, Technical University of Darmstadt, Peter-Grünberg-Str. 4, 64287 Darmstadt, Germany; 2Centre of Synthetic Biology, Technical University of Darmstadt, 64283 Darmstadt, Germany

**Keywords:** NK cell engager, checkpoint inhibitor, apoptosis, trispecific antibody, bispecific antibody, chicken-derived, immunotherapy, monoclonal antibodies, yeast surface display, common light chain

## Abstract

Valued for their ability to rapidly kill multiple tumor cells in succession as well as their favorable safety profile, NK cells are of increasing interest in the field of immunotherapy. As their cytotoxic activity is controlled by a complex network of activating and inhibiting receptors, they offer a wide range of possible antigens to modulate their function by antibodies. In this work, we utilized our established common light chain (cLC)-based yeast surface display (YSD) screening procedure to isolate novel B7-H3 and TIGIT binding monoclonal antibodies. The chicken-derived antibodies showed single- to low-double-digit nanomolar affinities and were combined with a previously published CD16-binding Fab in a 2+1 format to generate a potent NK engaging molecule. In a straightforward, easily adjustable apoptosis assay, the construct B7-H3xCD16xTIGIT showed potent apoptosis induction in cancer cells. These results showcase the potential of the TIGIT NK checkpoint in combination with activating receptors to achieve increased cytotoxic activity.

## 1. Introduction

Natural killer (NK) cells are innate lymphoid cells that possess the ability to rapidly identify and kill tumor cells. NK cells patrol the body in search of foreign pathogens or malignant cells, also called immunosurveillance [1]. Their activation is based on an array of activating and inhibiting receptors interacting with the surface of healthy or malignant cells [2]. While interacting with another cell, either the inhibitory signals prevail and the target cell is left unscathed or the activating signals take over, leading to the elimination of the target cell [3]. In particular, antibody-decorated cells are recognized by interaction with the CD16a receptor on the surface on NK cells and trigger potent antibody-dependent cellular cytotoxicity (ADCC) [4].

In order to utilize NK cells for tumor immunotherapy, NK cell engaging molecules, which combine tumor and NK cell binding moieties, can be used. These molecules facilitate the formation of an immunological synapse, promoting the cytotoxic activity of NK cells. Upon activation, NK cells express death ligands and exocytose cytotoxic granules containing perforin and granzyme [3]. Binding moieties can, among others, be in the form of single-chain variable fragments (scFv) [5], single-domain antibodies (sdAbs) [6] or classical antigen-binding regions (Fabs) [7]. Additionally, cytokines like IL-15 [8] or IL-2 [5] can be incorporated into the engager molecules, further activating the NK cell. Even combinations of IL-15 with the monomeric sushi domain of the IL-15 receptor have been developed for incorporation in NK cell-engaging molecules [9]. 

In NK immunotherapy, tumor targeting usually focuses on proteins preferentially expressed on tumorous tissues, including EGFR [10], BCMA [11], HER2 [12], and B7-H3 [13]. NK cell-targeting molecules can either engage activating receptors like CD16a [10,12,14] and NKp46 [15] or block inhibiting receptors, including NKG2A in the case of monalizumab [16] and TIGIT in the form of the monoclonal antibody vibostolimab [17].

B7-H3 (also called CD276) is a type I transmembrane protein and a member of the B7 family of checkpoint molecules regulating the adaptive immune system. On an mRNA level, B7-H3 is ubiquitously found throughout the body. However, protein expression is tightly regulated in healthy tissue by posttranscriptional control mechanisms [17]. In tumorous tissues, B7-H3 expression levels are often elevated and associated with poor prognosis and tumor aggressiveness [18,19,20,21]. Vallera et al. recently demonstrated that NK cells can effectively be recruited for tumor immunotherapy using trispecific engager molecules called TriKEs. By combining B7-H3 and CD16-targeting moieties with IL-15 in a single molecule, effective tumor killing could be achieved in vitro and in vivo [13].

CD16 is the low-affinity receptor for IgG1 and IgG3 (also known as FcγRIII). On NK cells, CD16 is the most potent activating receptor and consequently a promising antigen for the NK cell targeting function of a NK engager molecule. Activation of CD16 triggers cytotoxic pathways, enhances proliferation, and induces cytokine release [22]. The NK cell is detached from the target cell by ADAM17-mediated shedding of CD16 [23,24]. However, the engager molecule would lose contact with the NK cell in the process. To improve therapeutic efficiency, a second NK cell-targeting moiety could be introduced into the molecule.

TIGIT (T cell immunoglobulin and ITIM domain) is an inhibitory receptor on the surface of NK cells and acts as an immune checkpoint. It is part of the DNAM-1/TIGIT/PVRIG/TACTILE axis and binds to CD155 (poliovirus receptor) and CD112 (nectin-2) [25]. Upon activation, TIGIT inhibits NK cell degranulation and cytokine production [26]. By blocking CD155–TIGIT interaction, NK cell inhibition is prevented, preserving the cytotoxic activity of NK cells. Several anti-TIGIT mAbs have gone into clinical trials, underlining the therapeutic interest in this checkpoint molecule [27,28,29]. Rousseau et al. provide an extensive overview of anti-TIGIT therapies in their recent review, underlining the increasing interest in this NK checkpoint molecule [30].

We recently showed that chicken-derived common light chain antibodies can be combined into trispecific NK-engaging molecules in a 2+1 format [7]. Here, we expand on the idea, introducing a novel tumor-targeting anti-B7-H3 Fab and NK checkpoint-inhibiting anti-TIGIT Fab, resulting in a novel dual NK cell-targeting natural killer engager antibody (Figure 1).

In comparison to the molecules in the clinic, this molecule not only blocks the TIGIT NK cell checkpoint but also stimulates ADCC though CD16 targeting. Combined with the anti-B7-H3 Fab, the construct B7-H3xCD16xTIGIT induces potent cytotoxic activity of NK cells against tumor cells.

## 2. Results

### 2.1. Anti-B7-H3 Fab: Immunization, Screening, and Characterization 

The B7-H3 extracellular domain for chicken immunization was produced in-house as C-terminal StrepTag II fusion (B7-H3-SII). The protein was expressed in Expi293F cells and purified via a StrepTactin XT 4Flow column according to the supplier’s instructions. Immunization was performed with B7-H3-SII. Titer determination revealed high titers against B7-H3-SII. Based on the isolated RNA of chicken spleen cells, VH sequences were amplified by PCR, and a yeast surface display (YSD) library of 6.5 × 10^8^ transformants was generated in *S. cerevisiae* EBY100 [31]. The plasmid encoding the common light chain dFEB1 [7] was transformed into *S. cerevisiae* BJ5464 and mated with the anti-B7-H3 HC library to generate diploid yeast presenting cLC Fab fragments. Library diversity was approximately 6.5 × 10^8^ clones.

Screening was performed by FACS against B7-H3-SII for the first two rounds using StrepMAB-AF488 for detection of antigen binding and against biotinylated B7-H3-SII for the following third and fourth round using streptavidin allophycocyanin for antigen detection. The decrease from 1 µM to 50 nM of antigen ensured enrichment of high-affinity binders (Appendix A).

After the last screening round, single clones were analyzed for B7H3-SII-Biotin binding using a flow cytometer (Appendix A). The VH of seven positive clones was sequenced, and the four most diverse VH sequences (clones 3, 6, 9, and 12) were reformatted into full-length chimeric G1M3 Fc allotype LALA IgG1 format using a Golden Gate-based approach, as published previously [32]. Notably, the LALA mutations in the Fc region render the Fc largely nonfunctional with respect to CD16 binding. All four antibodies were expressed in Expi293F cells and purified via a MabSelect PrismA column. 

Binding affinities were determined against B7-H3-SII via BLI or on Expi293F cells that naturally express B7-H3 on their surface. BLI-based affinities ranged from 1.4 nM to 12.6 nM (Figure 2a). On-cell measurements resulted in apparent K_D_ values ranging from 0.4 nM to 9.3 nM (Figure 2b). The antibodies did not bind B7-H3 negative NK-92 cells.

### 2.2. Anti-TIGIT Fab: Immunization, Screening, Characterization, and Humanization

The TIGIT extracellular domain for chicken immunization was produced in-house as C-terminal Fc fusion (TIGIT-Fc) containing a TEV protease cleavage site between the TIGIT extracellular domain and the human Fc part. The protein was expressed in Expi293F cells and purified via a MabSelect PrismA column according to the supplier’s instructions. Subsequently, the Fc part was cleaved by addition of TEV protease. TEV was removed by a HisTrap excel column and the Fc by a MabSelect PrismA column. The flow-through of both columns contained pure TIGIT without any tags or impurities. Titer determination after chicken immunization revealed moderate titers against TIGIT. Based on the isolated RNA of chicken spleen cells, VH sequences were amplified by PCR, and a YSD library of 1.65 × 10^8^ transformants was generated in *S. cerevisiae* EBY100. As for the generation of the B7-H3 binders, the cLC dFEB1 was transformed into *S. cerevisiae* BJ5464 and mated with the anti-TIGIT HC library to generate diploid yeast presenting cLC Fab fragments. Library diversity was approximately 1.65 × 10^8^ clones.

Screening was performed against biotinylated TIGIT in three rounds, with antigen concentration decreasing from 1 µM to 50 nM to ensure enrichment of high-affinity binders (Appendix A). In a second screening campaign, the addition of TIGIT ligand CD155 and sorting for non-binding of CD155 enriched Fabs that specifically block the TIGIT–CD155 interaction (Appendix A).

After the last screening round, single clones were analyzed for TIGIT binding by flow cytometry (Appendix A). The VH of six positive clones was sequenced, and the three most diverse VH sequences (clones 4, 6, and 20) were reformatted into full-length chimeric G1M3 Fc allotype LALA IgG1 format using a Golden Gate-based approach, as published previously [32]. All antibodies were expressed in Expi293F cells and purified via a MabSelect PrismA column. 

Binding affinities were determined against TIGIT via BLI or on NK-92 H6 cells that naturally express TIGIT on their surface. NK-92 H6 is a NK-92 cell line that was stably transfected with CD16^F158^. BLI-based affinities were determined to be 21 nM and 15 nM for clones 4 and 6, respectively, while clone 20 showed no binding to TIGIT (Figure 3a). On-cell measurements resulted in apparent K_D_ values of 5.9 nM for both clones 4 and 6 (Figure 3b). Clone 20 appeared to be sticking to cells and was excluded from further experiments. Clone 4 was used for further engineering. It did not bind to TIGIT-negative RT112 and Expi HEK293 cells.

### 2.3. Construction and Characterization of Trispecific Dual-NK Engaging Antibody

The architecture of the trispecific antibody was based on the previous results from Bogen et al. [7], with the most notably properties being the knobs-into-holes technology for heavy chain heterodimerization (Figure 1) and the positioning of the anti-CD16 Fab [7] at the inner position of the double-Fab arm. To this end, the CD16 binding module with affinities to CD16 of around 5.6 nM was used from previous work. This resulted in the construct B7-H3xCD16xTIGIT. Three types of control molecules were also generated: engagers with only one NK target (B7-H3xTIGIT and B7-H3xCD16), one-armed variants (oa B7-H3 and oa CD16xTIGIT), as well as a variant with effector-competent Fc, also allowing the binding of all three targets simultaneously (B7-H3xTIGIT Fc comp.). All antibody constructs were assembled in the aforementioned format on a G1M3 Fc, which was either effector-silent (LALA) or effector-competent. Purification was carried out in a three-step approach, utilizing HisTrap excel, StrepTactin XT 4 Flow, and HiTrap Desalting columns, eliminating homodimers in the process. SDS-PAGE revealed the expected sizes for the different chains and high purity (Figure 4a). All six antibody constructs also showed high monomeric content in SEC (Figure 4b). 

On-cell affinities were measured on B7-H3-expressing RT112 bladder cancer cells or TIGIT- and CD16^F158^-expressing NK-92 H6 cells. All constructs containing the anti-B7-H3 Fab showed affinities of between 13 and 17 nM on RT112 cells (Figure 5a). On NK-92 H6 cells, affinities ranged from 26 nM to 89 nM (Figure 5b). Simultaneous binding of all Fabs built into the constructs was validated via BLI (Figure 5c). The antibody was immobilized on the biosensor surface and sequentially incubated with all three proteins of interest. Antigens were added one at a time in a consecutive manner.

### 2.4. Cell-Based NK Killing Assay

To assess the NK engaging capabilities of the constructs, an NK killing assay was performed. RT112 cells were Calcein-stained and co-incubated with NK-92 H6 cells at a 1:1 ratio either in presence or absence of antibodies, while untreated RT112 cells served as the negative control. As a positive control, maximum stimulation of NK degranulation was achieved by addition of PMA and ionomycin [33]. After a 3 h incubation period, cells were stained with AF647-conjugated Annexin V to measure the fraction of early apoptotic RT112 cells in the samples (Appendix A). B7-H3xCD16xTIGIT achieved killing comparable to the positive control, while B7-H3xTIGIT Fc comp. and oa TIGITxCD16 induced apoptosis to a smaller extent. B7-H3xCD16 showed weak induction of apoptosis, while B7-H3xTIGIT and oa B7-H3 treated samples showed apoptosis rates comparable to the PBS control. RT112 in the absence of any NK cells showed the lowest apoptosis rate (Figure 6a).

In order to better showcase the difference of CD16 engagement of Fabs compared to a competent Fc, the EC_50_ of B7-H3xCD16xTIGIT and B7-H3xTIGIT Fc comp. was determined. This resulted in EC_50_ values of 2 nM for B7-H3xCD16xTIGIT and 8 nM for B7-H3xTIGIT Fc comp (Figure 6b). The effector silent trispecific construct achieved around 30% maximum apoptosis induction, comparable to the induction achieved with PMA/ionomycin stimulation, while the effector-competent construct only achieved 15%.

### 2.5. PBMC Killing Assay

Moving from assays based on stable cell lines to human donor-derived material, we measured EC_50_ values in a PBMC-derived NK-based cell killing assay. For this, patient-derived NK cells were isolated from blood samples and rested overnight, before tumor cell killing was assessed in a flow cytometry-based assay. Target cells were fluorescently labelled with CellTracker Deep Red and incubated with isolated NK cells in an effector to target a cell ratio of 5:1 for four hours; viability was measured via staining with SYTOX Green. Antibodies were added in a 1:10 dilution series, with Cetuximab serving as a positive control and an effector silent one-armed counterpart (oa_hu225 (eff-)) serving as a negative control (Figure 7). To prove the robustness of our findings, experiments were conducted with the previously used RT112 cells as well as A549 cells.

Cetuximab achieved potent tumor cell killing, with EC_50_ values in the low picomolar range, followed by B7-H3xTIGITxCD16 with an EC_50_ of around 10–20 pM. The one-armed cetuximab counterpart oa_hu225, as reference for an antibody with just one tumor engaging Fab, also achieved two-digit nanomolar values, while the Fc-competent variant B7-H3xTIGIT performed in the low nanomolar range. 

To summarize, we generated a novel trispecific antibody combining tumor targeting, CD16 engagement, and TIGIT checkpoint inhibition in one molecule. The already published CD16 engager NKE14 [7] was combined with a chicken-derived anti-B7-H3 and anti-TIGIT Fabs in a cLC and effector silent knobs-into-holes format. This antibody showed apoptosis induction comparable to PMA/ionomycin stimulation in the NK-92-based assay and a slightly lower EC_50_ but high tumor cell killing compared to its effector-competent counterpart. In the PBMC-based assay, the difference in EC_50_ between the two constructs was 40–100-fold, depending on the target cell line.

## 3. Discussion

In this study, we presented a novel trispecific NK cell-engaging antibody B7-H3xTIGITxCD16, which can simultaneously activate NK cells through CD16 binding, block the inhibiting NK checkpoint molecule TIGIT, and redirect NK cells towards B7-H3-positive tumor cells. The molecule was constructed with a 2+1 architecture comprising the TIGIT- and CD16-binding Fab on the one side and the B7-H3-binding Fab on the other side. Its common light chain (cLC) architecture avoids LC mispairing, and incorporation of the knobs-into-holes technology increases the yield of heterodimeric protein. 

Regardless of the exact molecular architecture, the incorporation of a CD16 binding moiety in NK cell engager molecules is an established way to achieve NK cell activation [7,13,15,34,35,36,37]. Compared to an effector-competent Fc with an affinity to CD16 of around 1 µM, Fab fragments are able to bind to CD16, with a K_D_ several orders of magnitude lower. This increased affinity in turn leads to stronger ADCC [38]. As demonstrated in the killing assays, the trispecific construct B7-H3xTIGITxCD16 showed stronger apoptosis induction in target cancer cells and a lower EC_50_ compared to its Fc-competent counterpart. Our findings were robust and applicable to different types of effector cells (NK-92 H6 and patient-derived NKs) as well as target cell lines (RT112 and A549).

To retain homeostasis, activated CD16 is shed from the cell surface via ADAM17 [23,24], leading to a potential loss of contact of the antibody with the NK cell. To avoid a loss of contact, we incorporated a TIGIT-binding Fab fragment into our molecule. The NK checkpoint molecule TIGIT naturally binds to CD155 and CD112 and inhibits degranulation and cytokine production upon activation [26]. By binding to TIGIT and blocking this interaction, the inhibiting signaling is prevented, and the NK cell retains its activity. Simultaneously, the NK cell does not lose contact to the antibody in the case ADAM17 that cleaves activated CD16 from the cell surface. Upon synapse formation, the NK cell reduces CD16 expression and also sheds the protein from its cell surface. Although our antibody could still maintain NK cell contact with the TIGIT Fab, this engagement is not sufficient for strong NK cell activation (Figure 6a). Notably, no NK cell exhaustion was observed for another trifunctional NK cell engager simultaneously targeting CD16 with high affinity and NKp46 [39]. However, this may depend on the nature of receptor combinations and the affinities of the binding modules. Hence, the long-term impact of continuous NK cell activation and the potential for NK cell exhaustion or other adverse effects of a CD16xTIGIT NK cell engager remain to be elucidated. 

The synergistic effect of simultaneous CD16 activation and TIGIT checkpoint inhibition can be seen in the killing assays. In the case of bispecific antibodies, TIGIT inhibition alone has no effect on NK cell-mediated cell killing, and CD16 activation slightly increases cytotoxicity; the combination of both Fabs in one molecule, even without tumor targeting moiety, further increases the activating potential of the antibody. To our knowledge, incorporation of a TIGIT-blocking moiety into high-affinity CD16 targeting NK cell engager constructs has not yet been published, and this work underlines the potential of future efforts to target this inhibiting receptor in NK-based immunotherapy.

In order to effectively redirect NK cells to tumor cells, we isolated a novel B7-H3-binding Fab. B7-H3 mRNA is ubiquitously found throughout the body [17], but its limited expression in healthy tissue [18,19,20,21] makes it a promising target for immunotherapy. Dong et al. highlighted B7-H3 overexpression across various cancer types (bladder, breast, cervical, colorectal, esophageal, kidney, liver, lung, ovarian, pancreatic, prostate, glioma, and melanoma). Yang et al. and Picarda et al. summarized the fraction of cases that were B7-H3-positive for different cancer types [18,20,21]. More than 50 % of breast cancers display B7-H3 overexpression, and in colorectal carcinoma, more than 95 % of patient tumors were B7-H3 positive and B7-H3 expression was negatively associated with overall survival rate [40]. However, only a few examples utilizing B7-H3-binding moieties to redirect T or NK cells towards tumor cells can be found in the literature Among them, T cell engagers, CAR T cells, and a trispecific construct (TriKE) containing a B7-H3 scFv, an anti CD16 VHH, and rhIL15 as a proliferative signal to NK cells were described [6,13,41,42,43]. It remains to be established how our trispecific construct, which contains both a CD16 and a TIGIT Fab arm for NK cell activation, compares to these reported approaches, including consideration of combination therapies. Our data indicate that B7-H3 binding in a classical Fc non-silenced antibody format showed low impact on the viability of tumor cells; only the addition of a CD16 or both a CD16 and TIGIT Fab led to the increased cytotoxic potential of NK cells, corroborating the notion that more complex formats with carefully adjusted tumor and NK cell binding modules are required for a strong anti-tumor activity. 

In recent years, several formats for NK cell engagers have been developed, and a plethora of NK cell target combinations are currently being explored. These include FcγRIII binding but also the targeting of activating receptors NKG2D, NKp30, or NKp46 alone or in combination with CD16 binding. NK cell target combinations in trispecifics that simultaneously address two or more activating receptors or one activating receptor in combination with checkpoint inhibitors (TIGIT, NKGA, TIM3, LAG3) are less explored [44]. An Fc-competent PD-L1*TIGIT bispecific antibody promoted greater T cell expansion and tumor cell killing than benchmark antibodies and antibody combinations in an Fc-dependent manner, indicating that the TIGIT-mediated beneficial properties may be related to both T cell and NK cell activation [45].

The developability of multispecific antibodies is an issue to be considered early on, since the generation of more demanding formats may result in reduced stability, the formation of aggregates, and undesired pharmacokinetic profiles [46]. In a therapeutic setting, the main advantage of the IgG-based molecule compared to scFv-based variants like BiKEs and TriKEs [47] is the increased half-life of full-length antibodies through neonatal Fc receptor (FcRn)-mediated recycling. While proteins not interacting with FcRn may remain in the bloodstream for a few days without engineering its half-life, IgGs reportedly remain in the bloodstream for 19–21 days [48]. Moreover, the common light chain format used here contributes to both antibody stability and reduced complexity of the production of heterodimeric proteins. Of course, for further consideration in a therapeutic setting, the trispecific antibody would require humanization [49] and in-depth analysis of PK properties and safety profiles in animal studies.

In conclusion, the trispecific B7-H3xCD16xTIGIT NK cell engager combines tumor cell targeting by B7-H3 binding, NK checkpoint inhibition by TIGIT binding, and NK cell activation by CD16 binding in one molecule. Interestingly, the combination of CD16 and TIGIT binding in one molecule showed a synergistic effect regarding NK cell activation, which highlights the potential of TIGIT as a promising NK cell checkpoint molecule for further research. Utilization of a CD16-binding Fab compared to an Fc-competent molecule for CD16 engagement led to a slightly increased EC_50_ but drastically improved apoptosis induction in killing assays. Additionally, we established a straightforward method of measuring NK cell-induced apoptosis of tumor cells in a flow cytometry-based assay using a 1:1 ratio of target to effector cells and only 3 h of incubation time. 

Future studies may focus on the role of TIGIT in the context of CD16-based NK cell activation to further characterize the observed synergistic effect, particularly in comparison to other checkpoint molecules like PD-1. Furthermore, TIGIT blocking may also be used in combination with other NK-activating receptors like NKG2D or NKp30 to uncover previously hidden connections between NK cell signaling pathways. In order to translate our in vitro findings into therapy, future in vivo studies are required to validate our observations in a more complex biological environment. Based on the in vivo results, subsequent research focusing on pharmacokinetics, potential immunogenicity of the molecule in the human body, and safety considerations may enable novel treatment options in human immunotherapy.

## 4. Materials and Methods

### 4.1. Chicken Immunization

Two chickens (*Gallus gallus domesticus*) were immunized for each target protein. The immunization cocktails contained the Strep-tagged extracellular domain of B7-H3 and the extracellular domain of TIGIT (produced as TIGIT-Fc and purified after TEV cleavage). All proteins were expressed and purified in-house. All chickens received five immunizations on days 1, 14, 28, 42, and 56. At day 35, the serum titer against the target antigens was determined. The animal was sacrificed on day 63, followed by spleen resection and RNA extraction. The whole immunization procedure up to RNA extraction was performed by Davids Biotech GmbH (Regensburg, Germany). Experimental procedures and animal care were in accordance with EU animal welfare protection laws and regulations.

### 4.2. Yeast Strains and Plasmids

A pYD1-derived vector (Yeast Display Vector Kit, version D, #V835-01, Thermo Fisher Scientific, Waltham, MA, USA) was used for yeast surface display. The vector for the heavy chain library encodes the genes for the AGA2 signal peptide, followed by an MCS, the CH1 domain of human IgG1, the AGA2 gene, as well as a tryptophan auxotrophic marker and ampicillin resistance. Gene expression of the AGA2 fusion protein is controlled via a GAL1 promotor.

The common light chain-carrying plasmid is also pYD1-derived, coding for a αMFpp8 signal sequence followed by either the chicken (dFEB1 for screening) or human (hdFEB4-1-4 for humanization) VL, followed by a human lambda CL, and is also under the control of a GAL1 promotor. Both VL domains have been characterized and published by our group [7]. The plasmid also contains genes for a leucin auxotrophic marker and kanamycin resistance.

Saccharomyces cerevisiae strains EBY100 [MATa URA3-52 trp1 leu2Δ1 his3Δ200 pep4::HIS3 prb1Δ1.6R can1 GAL (pIU211:URA3)] (Thermo Fisher Scientific) or BJ5464 [MATα URA3-52 trp1 leu2Δ1his3Δ200 pep4::HIS3 prb1Δ1.6R can1 GAL] (American Type Culture Collection, Manassas, VA, USA) were cultured in YPD (20 g/L peptone/casein, 20 g/L glucose, and 10 g/L yeast extract). After transformation with the heavy chain (EBY100)- or light chain (BJ5464)-containing plasmids, cells were cultured in SD medium (5.4 g/L Na_2_HPO_4_, 8.6 g/L NaH_2_PO_4_ × H_2_O, 20 g/L glucose, 5 g/L ammonium sulfate, 1.7 g/L yeast nitrogen base (without amino acids)) supplemented with the amino acid mix appropriate for the auxotrophic marker encoded on the plasmid. Diploid yeast cells were cultured in SD-Trp-Leu medium for growth or SG-Trp-Leu for induction of surface presentation (same composition as SD medium, but with Galactose instead of Glucose).

### 4.3. Yeast Library Generation

The RNA of 10^8^ spleen cells was washed twice with 70% isopropanol, and the pellet was dried at 65 °C to evaporate the residual solvent. The pellet was resuspended in 50 µL DEPC-treated water (Thermo Fisher Scientific), and cDNA synthesis was carried out with the SuperScript III First-Strand Synthesis kit (Thermo Fisher) according to the suppliers’ instructions. One reaction contained 25 µL RNA, 5 µL Hexamer Primer, 5 µL dNTPs, 15 µL H_2_O, 10 µL 10× RT buffer, 20 µL 25 mM MgCl_2_, 10 µL 0.1 mM DTT, 5 µL RNAse Out, and 5 µL SuperScript III. Four reactions were carried out for each chicken, and the cDNA of all 8 reactions was pooled for each immunization campaign. Amplification of VH sequences was carried out using OneTaq polymerase (NEB) in a final volume of 50 µL. The primer pair used was VH NheI for/VH NotI rev. Each reaction contained 2 µL cDNA as the template. PCR was performed with the following protocol: 30 s at 94 °C initial denaturation, 33 cycles of 20 s at 94 °C, 30 s at 63 °C, and 30 s at 68 °C. Final elongation was performed for 5 min at 68 °C. Four reactions were pooled and purified via Wizard SV Gel and PCR Clean-up System (Promega, Madison, WI, USA). The CH1-carrying vector was linearized with NheI, NotI, and EcoRI (NEB) and purified via Wizard SV Gel and PCR Clean-up System. Library generation was carried out with *S. cerevisiae* strain EBY100, 3 µg linearized vector, and 10 µg VH insert, as specified in [50]. Transformed cells were cultured in SD-Trp medium.

*S. cerevisiae* strain BJ5464 was transformed with the plasmid coding for the dFEB1 cLC, cultured in SD-Leu, and used for mating to generate diploid yeast. Equal amounts of heavy chain- and light chain-carrying cells were mixed, centrifuged, and resuspended in YPD to achieve a final concentration of 2 × 10^8^ cells/50 µL. Cells were added dropwise (50 µL) to pre-warmed YPD agar plates and incubated at 30 °C overnight. The following day, cells were washed off the agar plate and cultured in SD-Trp-Leu medium.

### 4.4. Yeast Library Screening Procedure

Cells of the diploid yeast library were used to inoculate SG-Trp-Leu medium at an OD_600_ of 1 and were incubated overnight at 30 °C and 120 rpm. The next day, cells were harvested by centrifugation and washed once with PBS-B (PBS + 0.1% (*w*/*v*) BSA) before resuspension in buffer with the desired concentration of the target antigen. After 30 min incubation on ice, the cells were washed twice with PBS-B and resuspended in a 1:75 dilution of either goat anti-human-Lambda Alexa Fluor 647 F(ab’)2 antibody (SouthernBiotech, Birmingham, AL, USA) and StrepMAB-Immo DY-488 conjugate (IBA Lifesciences, Göttingen, Germany) or goat anti-human-Lambda-PE F(ab’)2 antibody (SouthernBiotech) and Strepavidin-APC (Invitrogen, Waltham, MA, USA). After 15 min incubation on ice in the dark, cells were washed twice with PBS-B and screened by FACS using a Sony SH800S.

### 4.5. Reformatting, Expression, and Purification of Full-Length Antibodies

Plasmids were isolated from yeast cells using Zymoprep Yeast Plasmid Miniprep Kit I (Zymo Research, Irvine, CA, USA) according to the suppliers’ instructions. Plasmids were transformed into *E. coli* XL1blue, and VH domains were sequenced by Microsynth Seqlab (Göttingen, Germany). Sequences chosen for reformation were amplified using Q5 Polymerase (NEB) according to the suppliers’ instructions. The utilized primers incorporated terminal SapI sites for subsequent Golden Gate cloning. The Golden Gate reaction was performed using a pTT5-derived (Expresso CMV-based system, Lucigen, Middleton, WI, USA) destination vector, an entry vector carrying human IgG1 CH1-CH2-CH3, and the generated PCR product as well as SapI and T4 ligase (NEB), according to the suppliers’ instructions. *E. coli*. XL1blue was transformed with the reaction mixture, and sequences were validated by Sanger Sequencing. 

Protein expression was carried out in Expi293F cells (Thermo Fisher), which were cultivated in Expi293 Expression Medium (Gibco, Grand Island, NY, USA) at 37 °C, 8% CO_2_, and 110 rpm in a humidified atmosphere. Transient transfection was performed with Transporter 5 (Polysciences, Warrington, PA, USA) and 2 plasmids in a 1:1 mass ratio of heavy chain–light chain. For heterodimeric antibodies a 1:1:1 ratio of the three plasmids was used. 

Five days after transfection, the supernatant was sterile-filtered and purified using an Äkta pure system (Cytiva, Marlborough, MA, USA). Homodimeric antibodies were purified using a MabSelect PrismA column (Cytiva), with subsequent buffer exchange via a HiTrap Desalting column (Cytiva). Heterodimeric antibodies were purified with HisTrap Excel (Cytiva) and StrepTactin XT 4Flow (IBA Lifescience) columns, followed by buffer exchange via HiTrap Desalting column. All chromatographic steps were performed according to the manufacturers’ instructions.

### 4.6. SDS-PAGE

To characterize the expressed antibody constructs, SDS-PAGE was performed. For this, 4 µg of purified antibody was loaded onto Mini-PROTEAN TGX 4–15% gel (BioRad, Hercules, CA, USA) with reducing Laemmli buffer and subsequently stained with colloidal Coomassie [51].

### 4.7. Affinity Determination and Simultaneous Binding Assay via Biolayer Interferometry

For affinity determination, anti-human IgG-Fc capture (AHC, Sartorius, Göttingen, Germany) biosensors were equilibrated in PBS for 10 min and loaded with 10 µg/mL antibody solution until a layer thickness of 1 nm was reached. After a 30 s quenching step in kinetics buffer (KB, Sartorius), association was measured for 300 s against varying antigen concentrations diluted in KB. All antigens were produced in-house (B7-H3 with C-terminal StrepII-Tag, TIGIT and CD16 with N-terminal His_6_ and C-terminal TwinStrep-Tag), and analyzed concentrations ranged from 500 nM to 7.8 nM. Subsequent dissociation in KB was measured for 300 s. KB without antigen was used as a negative control. Binding kinetics were calculated based on Savitzky-Golay filtering and a 1:1 Langmuir binding model.

For the simultaneous binding assay, AHC biosensors were equilibrated in PBS for 10 min and loaded with 10 µg/mL antibody until a layer thickness of 1 nm was reached. After a 30 s quenching step in PBS, association was measured sequentially for a 250 nM solution of all antigens. As a control, the association of only one or two antigens with subsequent incubation in PBS was also measured. The antigens used were B7-H3-SII, TIGIT, and His-CD16-TwinStrep.

All measurements were performed using the Octet RED96 system (ForteBio, Fremont, CA, USA) at 30 °C and 1000 rpm.

### 4.8. Size Exclusion Chromatography

Size exclusion chromatography (SEC) was performed using a TSKgel SuperSW3000 column (Tosoh Bioscience, Griesheim, Germany) together with a 1260 Infinity chromatography system (Agilent Technologies, Santa Clara, CA, USA). Chromatography was performed at a flow rate of 0.35 mL/min for 20 min, and absorbance at 280 nm was measured to detect protein elution. 

### 4.9. Cultivation of NK-92 H6 Cells

NK-92 H6 is a modified human NK-92 cell line (DSMZ ACC 488) that was stably transfected with CD16^F158^ in-house. Culture medium consisted of RPMI-1640 (Gibco) supplemented with 20% fetal bovine serum (FBS) superior (Merck Millipore, Burlington, MA, USA), 1% Penicillin/Streptomycin solution (Sigma Aldrich, St. Louis, MO, USA), 250 µg/mL G418 (Sigma Aldrich), and 10 ng/mL hIL-15 (produced in-house). Cells were cultured at 37 °C and 5% CO_2_ under a humidified atmosphere in a T75 flask and passaged every two to three days.

### 4.10. Cultivation of RT112 and A549 Cells

RT112 (ACC 418) and A549 (ACC 107) cell lines were obtained from DSMZ (German Collection of Microorganisms and Cell Cultures GmbH, Braunschweig, Germany). RT112 culture medium consisted of RPMI-1640 (Thermo Fisher Scientific) supplemented with 10% fetal bovine serum (FBS) superior (Merck Millipore) and 1% Penicillin/Streptomycin solution (Sigma Aldrich). A549 was maintained in Dulbecco’s Minimal Eagle Medium (DMEM) supplemented with 10% FBS and 1% Penicillin/Streptomycin solution (Sigma Aldrich). Cells were cultured at 37 °C and 5% CO_2_ under a humidified atmosphere in a T75 flask and passaged every three to four days after reaching 80% confluence.

### 4.11. On-Cell Binding Affinities

Binding to target and effector cells was determined by affinity titration. The cell lines used were the B7-H3-positive RT112 bladder cancer cell line or CD16/TIGIT double-positive NK-92 H6 cells. For staining, 10^5^ cells/well were seeded in 96-well plates, washed with PBS-B, and incubated with varying concentrations of the respective antibody. After 30 min on ice, cells were washed twice with PBS-B and incubated with a 1:75 dilution of PE-conjugated anti-human IgG–Fc for 15 min in the dark. After two washing steps, mean fluorescence of whole cell population was measured using a CytoFLEX S (Beckmann Coulter, Brea, CA, USA). Data points were plotted against logarithmic antibody concentration, and curves were fitted with a variable-slope four-parameter fit using GraphPad Prism. All measurements were performed in triplicate, and experiments were repeated at least 3 times, yielding comparable results.

### 4.12. NK-92 Killing Assay and EC_50_ Determination

RT112 cells were washed twice with PBS and treated with Trypsin/EDTA (Gibco) solution until cells detached from the culture flask. Afterwards, cells were collected by centrifugation at 500× *g* for 4 min and resuspended in fresh PBS. CalceinAM (Invitrogen) was added to a final concentration of 100 nM, and cells were incubated for 20 min at 37 °C and 5% CO_2_ in a humidified atmosphere. Subsequently, cells were collected and resuspended in RPMI + 10% FBS. At the same time, NK-92 H6 cells were harvested by centrifugation and washed with PBS before resuspension in RPMI + 10% FBS. Both cell suspensions were adjusted to a density of 5 × 10^5^ cells/mL, and 100 µL of each solution was mixed in a round-bottom 96-well plate, resulting in a 1:1 ratio of fluorescently labelled target and effector cells. A total of 200 µL of each cell suspension alone served as the negative control. A total of 20 µL of a 10 µg/mL solution of PMA/ionomycin in PBS was added to the positive control. Antibodies were diluted in PBS, and 20 µL of each solution was added to the well. PBS was added to the untreated control and negative controls. The plate was incubated for 3 h at 37 °C and 5% CO_2_ in a humidified atmosphere. All experiments were performed in triplicate.

To analyze apoptosis induction, cells were collected by centrifugation for 3 min at 800× *g* and thoroughly resuspended in 25 µL Annexin V staining buffer (10 mM HEPES, 140 mM NaCl, and 2.5 mM CaCl_2_, pH 7.4) containing 1 drop Annexin V Ready Flow AF647 reagent (Invitrogen)/500 µL buffer. Cells were incubated for 15 min at room temperature, and 100 µL Annexin V staining buffer was added to dilute the samples before analysis using a CytoFlex S. Calcein-positive RT112 cells were gated in the FITC channel, while AF647 fluorescence was analyzed in the APC channel. Apoptosis induction was measured by dividing the population into AF647 high- and AF647 low-signal subsets. The ratio of AF647 high cells was plotted using GraphPad Prism.

### 4.13. PBMC Killing Assay and EC_50_ Determination

PBMCs were isolated from whole blood samples of healthy donors by density gradient centrifugation using SepMate™ PBMC Isolation Tubes (StemCell Technologies, Vancouver, BC, Canada). NK cells were then isolated with the EasySep™ Human NK Cell Isolation Kit (Stemcell Technologies) and rested overnight at 1.2 × 106 vc/mL in AIMV media (Thermo Fisher Scientific) containing 100 U/mL recombinant human interleukin-2 (Acro Biosystems, Newark, DE, USA). RT112 and A549 cells were prepared the next day by parallel staining with CellTracker™ Deep Red Dye (Thermo Fisher) according to the manufacturer’s instructions. The target cells were seeded in a 384-well clear-bottom microtiter plate (Greiner Bio-One, Kremsmünster, Austria) with 2500 cells/well in 17.5 µL. After 3 h of incubation at 37 °C, 10 µL of the samples was added at 200 nM in a 1:10 dilution series, followed by the addition of 17.5 µL NK cells in an effector cell to target a cell ratio of 5:1. SYTOX™ Green Dead Cell Stain (Invitrogen) was added to the plate for a final concentration of 0.03 μM. Effector cells cultivated with target cells without antibodies, as well as a monovalent EGFR targeting Fc effector-silenced antibody derivative (oa_hu225 (eff-)), were used as negative controls. To analyze tumor cell lysis, the overlay signals (green and red fluorescence) displayed dead target cells only, which were evaluated after 4 h using the Incucyte^®^ Live Cell Analysis System (Sartorius). Target cell lysis is displayed after basal killing subtraction.

## Figures and Tables

**Figure 1 molecules-29-01140-f001:**
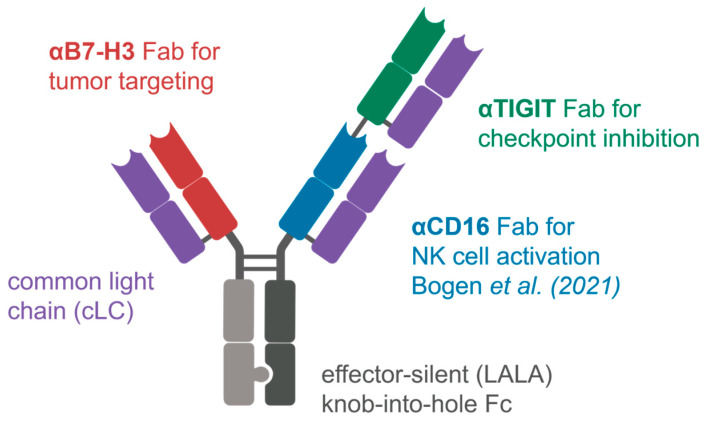
Schematic representation of the trispecific NK engager B7-H3xCD16xTIGIT. The anti-CD16 Fab was previously published by our group [7].

**Figure 2 molecules-29-01140-f002:**
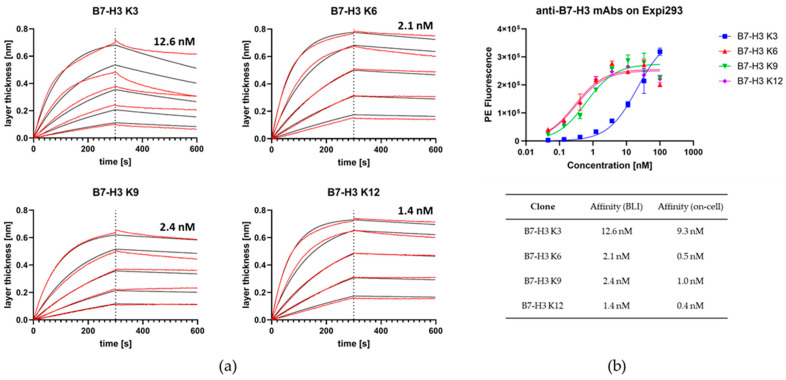
Affinity determination of anti-B7-H3 antibodies. (**a**) BLI-based affinity determination of anti-B7-H3 Fabs K3, K6, K9, and K12. Measurements were performed with antibodies in IgG1 format. Red lines indicate measured layer thickness, while black lines represent fit for affinity determination. (**b**) Affinity determination of anti-B7-H3 antibodies on B7-H3-expressing Expi293 cells.

**Figure 3 molecules-29-01140-f003:**
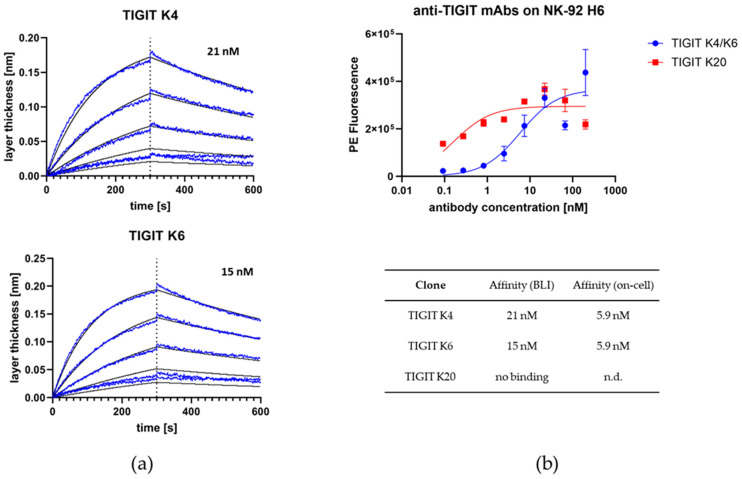
Affinity determination of anti-TIGIT antibodies. (**a**) BLI-based affinity determination of anti-TIGIT antibodies K4 and K6. No binding was observed for K20. Measurements were performed with antibodies in IgG1 format. Blue lines indicate measured layer thickness, while black lines represent fit for affinity determination. (**b**) Affinity determination of anti-TIGIT antibodies K4, K6, and K20 on TIGIT-expressing NK-92 H6 cells.

**Figure 4 molecules-29-01140-f004:**
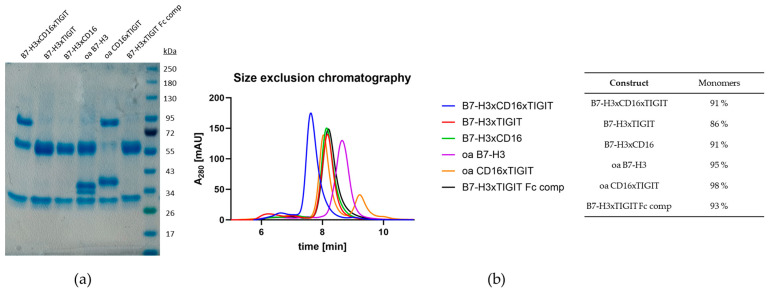
Biophysical characterization of the multispecific antibodies. (**a**) SDS-PAGE of the multispecific antibody constructs. (**b**) SEC chromatograms and monomeric content of multispecific antibody constructs.

**Figure 5 molecules-29-01140-f005:**
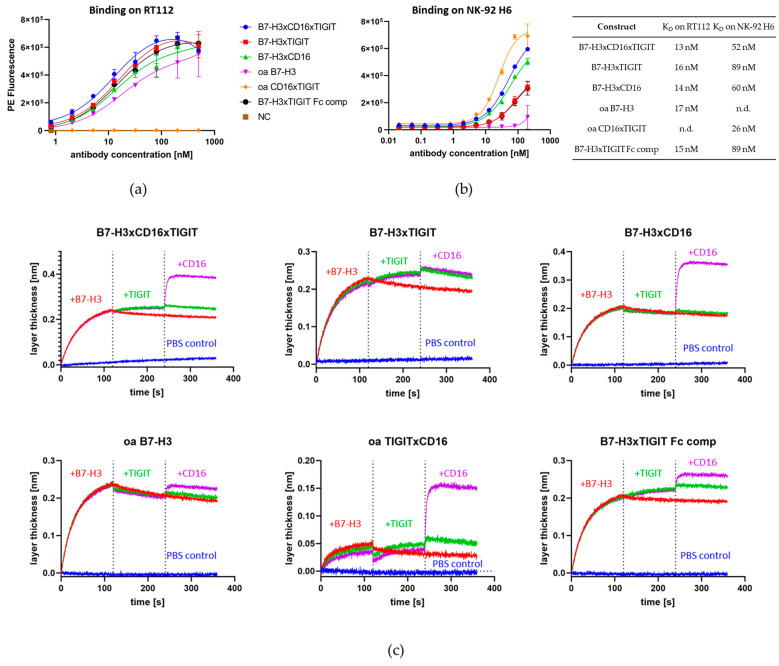
Analysis of on-cell binding affinities and simultaneous binding capabilities. (**a**) On-cell binding of multispecific antibodies on RT112 bladder cancer cells. (**b**) On-cell binding of multispecific antibodies on NK-92 H6 cells. (**c**) Simultaneous binding experiments measured via BLI. The antibody was immobilized on the biosensor and sequentially incubated with all three proteins of interest in a sequential manner, adding one antigen at a time.

**Figure 6 molecules-29-01140-f006:**
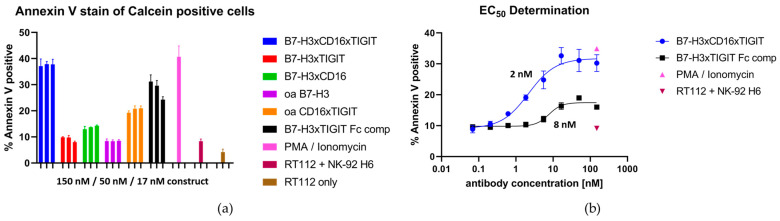
Killing assays with RT112 and NK-92 H6 cells. (**a**) Apoptosis induction capabilities of serial dilutions of the multispecific antibody constructs. (**b**) EC_50_ determination of the trispecific antibody constructs.

**Figure 7 molecules-29-01140-f007:**
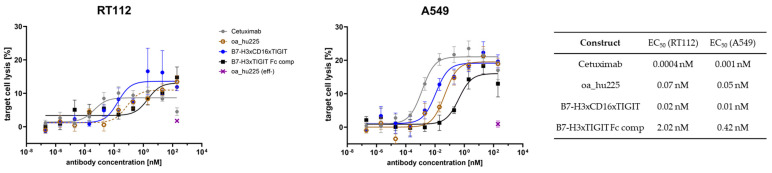
PBMC killing assay with RT112 (**left**) or A549 (**right**) as target cell line. Patient-derived NK cells from healthy donors were incubated with fluorescently labelled target cells in an effector to target ratio of 5:1 for 4 h. Antibody constructs were added as 1:10 dilution series starting at 200 nM. Cetuximab served as a positive control, and its one-armed effect silent counterpart (oa_hu225 (eff-)) as a negative control.

## Data Availability

The data presented in this study are available in article and Appendix A.

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
