# Peer review of "Potent Apoptosis Induction by a Novel Trispecific B7-H3xCD16xTIGIT 2+1 Common Light Chain Natural Killer Cell Engager"

_molecules, 2024, doi:10.3390/molecules29051140_

Round 1
Reviewer 1 Report
Comments and Suggestions for Authors
In this manuscript, Ulitzka et al. has isolated novel B7-H3 and TIGIT binding monoclonal antibodies, which were combined with the CD16-binding Fab to generate a potent NK engaging molecule. Importantly, the authors have demonstrated that the NK engaging molecule can potently induce apoptosis in cancer cells. Overall, the documentation of the methods is comprehensive and the data can support the conclusions. However, some statistical test is missing and additional experiments are required to further confirm the findings from the study.
Specific comments:
1) Why only one cell line RT112 has been used in experiments? At least another cancer cell line should be used to repeat the experiments and confirm the findings.
2) Figure 6a, in addition to various concentrations of construct, has multiple incubation time points been tested?
3) Figure 6a, statistical tests need to be performed to compare the percentage of Annexin V positive cells in different treatment groups.
Reviewer 2 Report
Comments and Suggestions for Authors
PFA
